# The Impact of Heavy Metal Accumulation on Some Physiological Parameters in *Silphium perfoliatum* L. Plants Grown in Hydroponic Systems

**DOI:** 10.3390/plants12081718

**Published:** 2023-04-20

**Authors:** Radu Liviu Sumalan, Vlad Nescu, Adina Berbecea, Renata Maria Sumalan, Manuela Crisan, Petru Negrea, Sorin Ciulca

**Affiliations:** 1Faculty of Engineering and Applied Technologies, University of Life Sciences “King Mihai I” from Timisoara, 119 Calea Aradului, 300645 Timisoara, Romania; nescuvlad@gmail.com (V.N.); renatasumalan@usab-tm.ro (R.M.S.); 2Faculty of Agriculture, 119 Calea Aradului, University of Life Sciences “King Mihai I” from Timisoara, 300645 Timisoara, Romania; adinaberbecea@yahoo.com; 3“Coriolan Dragulescu” Institute of Chemistry, 24 Mihai Viteazul Blvd., 300223 Timisoara, Romania; mdorosencu@yahoo.com; 4Faculty of Industrial Chemistry and Environmental Engineering, Politehnica University Timisoara, 6 Bv. Vasile Parvan, 300223 Timisoara, Romania; petru.negrea@upt.ro

**Keywords:** phytoremediation, bioaccumulation, proline, photosynthetic pigments, enzymatic activity

## Abstract

Heavy metals like cadmium (Cd), copper (Cu), lead (Pb), and zinc (Zn), resulting from anthropogenic activities, are elements with high persistence in nature, being able to accumulate in soils, water, and plants with significant impact to human and animal health. This study investigates the phytoremediation capacity of *Silphium perfoliatum* L. as a specific heavy metal hyperaccumulator and the effects of Cu, Zn, Cd, and Pb on some physiological and biochemical indices by growing plants under floating hydroponic systems in nutrient solutions under the presence of heavy metals. One-year-old plants of *S. perfoliatum* grown for 20 days in Hoagland solution with the addition of (ppm) Cu-400, Zn-1200, Cd-20, Pb-400, and Cu+Zn+Cd+Pb (400/1200/20/400) were investigated with respect to the control. The level of phytoremediation, manifested by the ability of heavy metal absorption and accumulation, was assessed. In addition, the impact of stress on the proline content, photosynthetic pigments, and enzymatic activity, as being key components of metabolism, was determined. The obtained results revealed a good absorption and selective accumulation capacity of *S. perfoliatum* plants for the studied heavy metals. Therefore, Cu and Zn mainly accumulate in the stems, Cd in the roots and stems, while Pb mainly accumulates in the roots. The proline tended to increase under stress conditions, depending on the pollutant and its concentration, with higher values in leaves and stems under the associated stress of the four metals and individually for Pb and Cd. In addition, the enzymatic activity recorded different values depending on the plant organ, its type, and the metal concentration on its substrate. The obtained results indicate a strong correlation between the metal type, concentration, and the mechanisms of absorption/accumulation of *S. perfoliatum* species, as well as the specific reactions of metabolic response.

## 1. Introduction

“Heavy metals” is a generic term that is used to define the metals and metalloids group with an atomic weight of over 4000 kg m^−3^ [1,2]. Heavy metals are naturally present in the environment but can become a problem when released in excessive amounts from natural and/or human activities.

Industrial activities, such as mining, fertilizer and pesticide production, as well as electronics manufacturing, have resulted in significant amounts of heavy metals in the environment over the last century [3,4]. A range of natural processes, including volcanic activity, corrosion, evaporation, erosion, and geological weathering can also increase heavy metal pollution in different ecosystems [5,6].

Heavy metal pollution and contamination have become major problems worldwide and have increased exponentially in recent years, particularly affecting the soil, water, and air.

Heavy metals present in terrestrial and aquatic ecosystems can cause serious health problems for humans and all living organisms because most of them cannot be degraded either chemically or microbiologically, so their storage in time and space is long [7]. The metal toxicity depends on the type, amount, and duration of its exposure.

In the atmosphere of urban and industrialized areas, heavy metals such as Pb, Zn, Cr, Ni, or Cd are released as a result of traffic emissions, incomplete combustion of fossil fuels, or industrial processes. The extent of pollution needs to be seen in the context of urban areas expanding, having a detrimental impact on the health of people.

The high level of persistence and the ability to be easily transferred between different environmental ecosystems make heavy metal pollution extremely difficult to control. The accumulation and transfer of heavy metals in various environmental ecosystems facilitate their penetration into the food chain and pose a significant threat to human health and environmental security.

Therefore, heavy metals and metalloids are major risk factors for human, animal, and plant health due to their persistence in the environment, toxicity, bioavailability, and bioaccumulation potential.

Heavy metals can be grouped into two classes depending on their toxicity, essential and non-essential.

(1) Essential heavy metals are harmless or relatively harmless at low concentrations (Zn, Cu, Fe, and Co).

(2) Non-essential metals are toxic, even at low concentrations (Cd, Pb, Hg, As, and Cr).

Although essential heavy metals are also classified as micronutrients playing important roles in physiological and biochemical processes in plants, at high concentrations in soils they usually have toxic effects on plants, e.g., low biomass accumulation, chlorosis, suspension of growth and photosynthesis, changes in water balance and nutrient absorption and even senescence, leading to plant death [8,9]. In humans, Cu and Zn are generally considered non-toxic metals, with toxic effects such as intravascular hemolysis, liver and kidney failure [10] in the case of Cu, and nausea, vomiting, epigastric pain, lethargy, and fatigue symptoms for Zn occurring extremely rarely [11].

Accumulation of heavy metals in soils also affects pH, making soil alkaline or acidic [12], and may contaminate food as a result of soil and root interaction. Non-essential heavy metals like Pb have a huge negative impact on human health, as they can remain in the soil for over 150 years [13]. In plants, Pb can have a detrimental effect at the morphological, physiological, and biochemical levels. For humans, Pb is considered carcinogenic (Group 2B) and is mainly absorbed through the ingestion of food and water. Up to 70% can be taken in by the body [14].

Cd is a highly mobile heavy metal and a dangerous element in soil because of its greater solubility in water [15]. Cd affects plant growth by altering nitrogen metabolism, membrane function, and chlorophyll biosynthesis [16]. Cd induces severe toxicity in various organs [17,18] and can lead to testicular and prostate diseases, as well as lung cancer, being classified as a type I carcinogen by the International Agency for Research on Cancer [19].

It is therefore essential to develop effective methods of eliminating these contaminants from polluted ecosystems to protect and maintain the natural balance of our environment. Over time, several technologies have been developed to reduce the effects of heavy metal pollution on ecosystems, such as chemical extraction, chlorination, electrokinetic adsorption, ion exchange, bioleaching, heat treatment, phytoremediation, and bioremediation.

Recent research has shown that combining multiple remediation techniques such as physical separation, isolation, immobilization, reducing toxicity, and extraction provides the best results. Thus, several technologies have been optimized: chemical-biological remediation provides an ecological and economically friendly solution for treating wastewater with heavy metals [20]; microbial-electrokinetic remediation produces useful bioelectricity and biofuel using microbial activity [21]. Electrokinetically enhanced phytoextraction is based on the removal of heavy metals from the soil by lowering the pH of the soil to 1.5, which increases the solubilization and bioavailability degree of heavy metals, followed by phytoextraction. Phytobial remediation is a process using plant species with bioaccumulation potential to extract heavy metals from waters and soils, and microorganisms for the degradation of metal compounds. Bacteria optimize plant growth by synthesizing substances like organic acids, deaminases, siderophores, and biosurfactants that convert metals into available forms [22].

Phytoremediation is the process by which plants effectively remove heavy metals by absorbing them from the contaminated substrate [23]. After absorption, they are either degraded or brought to less harmful forms or transported and accumulated in the various organs [24]. Phyto-stabilization and phytoextraction are the most widely used phytoremediation techniques. Plant species able to store significant heavy metal amounts in various organs (especially roots and shoots) are considered hyperaccumulators [25,26]. These plants must have a rapid growth rate with high biomass production potential, deep and branched root systems, and be tolerant to biotic and abiotic stressors [27]. Recommended plant species for heavy metal extraction shall provide translocation factor (TF) values and a bioaccumulation factor (BAF) above 1. The ideal candidates would be plant species with high tolerance and microbiological diversity associated with a low rate of metal transport [28].

*S. perfoliatum* L. (Asteraceae) is one of the indicated species for the remediation of ecosystems subject to heavy metal pollution [29] because it has an intense growth rate [30,31], it is tolerant to biotic and abiotic stressors [32,33], it produces a significant amount of vegetative biomass [34,35] that can be harvested and used in the production of biogas [36,37,38], and it has high environmental plasticity [39]. Previous studies have confirmed high values of BAF, TF, and removal efficiency (RE) values under the conditions of cultivation on soils polluted by heavy metals [39,40,41,42].

The most generalized effect of heavy metals in plants is their attack on the photosynthetic apparatus. This property is common to all heavy metals and is not specific to a particular metal, which makes measuring the photosynthetic activities a good screening method for detecting heavy metal stress [43,44].

The enzymatic system plays an essential role due to its antioxidant properties [45]. They can be used as sensitive biomarkers of different types of oxidative stress. The activity of enzymes can be disrupted when they are exposed to stress caused by heavy metals. The heavy metals bind to the enzymes mainly through electrostatic forces, resulting in significant disturbances in the metabolic processes of the enzymes [46].

High concentrations of heavy metals can lead to the formation of free radicals and reactive oxygen species, potentially resulting in oxidative stress. Proline (pro) accumulation is increasingly being recognized as an important indicator of oxidative stress by stabilizing the cell membrane. Pro is known as an osmo-protectant and a signaling molecule, as well as having an essential role in primary metabolism as free amino acids and as part of proteins [47]. Several types of research attest to a positive correlation between the accumulation of free pro and the increase of plants’ tolerance to stress factors [48,49,50]. In the presence of heavy metals, pro acts as a metal chelator, with the role of antioxidant defense but also a messenger [50].

Therefore, the research aimed at determining the phytoremediation capacity of the perennial herbaceous species *S. perfoliatum* and the effects of high concentrations of heavy metals on some physiological and biochemical parameters with implications on metabolic processes, by growing plants in floating hydroponic systems in nutrient solutions with an excess of heavy metals (Cu, Zn, Cd and Pb). Free proline (pro) and photosynthetic pigment contents (chlorophyll and carotenoids) under both individual and cumulative effects of four heavy metals were investigated. Furthermore, the impact of heavy metal accumulation on the enzymatic activities of superoxide dismutase (SOD), catalase (CAT), and peroxidase (POD) in different organs of *S. perfoliatum* plants was evaluated.

## 2. Results

### 2.1. Dynamics of Heavy Metal Accumulation in S. perfoliatum Plant Organs (Phytoextraction)

Appendix A data show the homogeneity of variance for heavy metal concentrations in different organs (roots, stems, and leaves) of *S. perfoliatum* plants grown under different treatments. Analysis of variance (ANOVA) performed for all heavy metals (Appendix A) revealed significant differences across plant organs.

Experiments on *S. perfoliatum* plants grown in Hoagland nutrient solution, without the addition of heavy metals, showed different responses regarding the accumulation of the four heavy metals (Cu, Zn, Cd, and Pb) in organs (Table 1). Thus, the Cu content in roots was superior to the other organs, on the background of a lower value in the leaves. The stems showed a significantly higher Zn content than the other two organs, which had close values. Cd accumulation was significantly higher in the leaves and lower in the stems. In the case of Pb, against the background of a higher concentration in the roots, there is a significant increase in leaves compared to the stems. The Cu (3.63–8.16 μg g^−1^) and especially Zn (23.50–39.83 μg g^−1^) contents accumulated by these plants were far superior to Cd and Pb.

Cu pollution of the substrate has caused accumulation of this element mainly in stems (810.78 μg g^−1^), and substantially equal values (333.81–360.94 μg g^−1^) in the other two organs. This treatment was also associated with a significant increase in Zn and Cd in the stems, and Pb in the roots, respectively. These plants recorded the lowest concentrations of Zn in the roots, Cd in the leaves, and Pb in the stems.

*S. perfoliatum* plants revealed similar responses to Zn pollution regarding Cu and Pb accumulation, with higher concentrations in roots and lower in stems. Zn was observed to have a significantly higher content of 120.6 μg g^−1^ in the roots, while the other organs had roughly equal values ranging from 73.43–75.78 μg g^−1^. Cd showed a higher amount in the root, equivalent to those accumulated in the stems and leaves.

In the Cd-polluted solution, the Cu content was higher in the roots compared to the other organs of the plants, while values were lower in the stem. Zn recorded a significantly higher concentration in the leaves and a reduced one in the roots, and Cd showed a superior accumulation of 155.18–159.89 μg g^−1^ in the roots and stems compared with a significantly lower value in the leaves. The Pb content in the roots and leaves of these plants was significantly higher than the determined values in the strains.

The high concentration of Pb in the nutrient solution increases the accumulation of the other three metals in the roots, while reducing the Cu concentration in the stems and Zn and Cd levels in the leaves. The obtained results indicate a Pb accumulation mainly in roots, compared with stems and leaves which have considerably lower values.

The combined effect of the four heavy metals generated similar outcomes on the accumulation of Cu and Cd, recording significantly different values in the organs, higher in stems and lower in the foliar apparatus. In addition, this treatment led to a significant increase in the amount of Zn in the stems, while the roots and leaves were found to have closely comparable values. Pb was mostly stored in the stems, with almost similar values in the other two organs.

Pb pollution had a clear impact on the amount of Cu and Zn in particular components of plants. An increase was observed in the amount of Cu in all plant organs, while roots saw an increase in Zn level. The excess of Cu led to an increase in Zn concentration in the stems and leaves. Additionally, Zn pollution induces a higher amount of Pb accumulation in plants, compared to values associated with Cu and Cd treatments.

Plants exposed to Pb stress had the highest amount of heavy metals detected, 2222–2503 µg g^−1^ while Zn and Cd pollution combined had a significantly lower amount, 536–578 µg g^−1^ (Figure 1). In the case of the control and in the pollution with Cu, a higher accumulation was found in stems (41.22–52.47%) with close values in roots (22.44–30.81%) and leaves (25.09–27.97%). Zn and Cd pollution caused the accumulation of heavy metals in roots (42.18–55.24%), with relatively balanced values in the other organs. By applying Pb pollution and the one with all four metals, plants have accumulated mainly in the root system (64.97–89.92%), compared to significantly lower amounts in the leaves (6.14–8.67%).

### 2.2. Evaluation of the Effects of Heavy Metal Exposure on the Free Proline and Assimilating Pigments Content

#### 2.2.1. The Free Proline Content

The pro content recorded values ranging from 1.91 mmol g^−1^ fw in control plants and 6.45 mmol g^−1^ fw in the case of plants stressed simultaneously by the four heavy metals (Figure 2). Under the effect of the induced stress by the cumulative action of heavy metals, a significantly higher amount of pro was synthesized in the plants of *S. perfoliatum* with 11.7–118.6% compared to the values related to individual stress. By Cd and Pb pollution, the pro amount synthesized was significantly higher by 166–188%, compared to the values recorded in the control. The addition of the individual polluting doses of Cu and Zn in the nutrient solution generated a small and insignificant variation in the amount of pro compared to the control.

#### 2.2.2. Photosynthetic Pigments

The information from Appendix A attests to the homogeneity of variance for photosynthetic pigments in *S. perfoliatum* plants grown under different metal types/concentrations. Results of ANOVA for photosynthetic pigments (Appendix A) revealed significant differences across plant organs.

Data analysis on photosynthetic pigments in *S. perfoliatum* grown in hydroponic media with different types of heavy metals shows that the total amount of chlorophyll (chl) recorded a reduced variation between 14.58 μg cm^−2^ for Cd pollution, and 21.06 μg cm^−2^ for Pb (Table 2). It is therefore noted that Cd pollution has led to a significant reduction in the amount of chlorophyll a (chl a), both in relation to the control and other metals.

Heavy metal pollution had a greater influence on the amount of chlorophyll b (chl b), with variations from 3.64 μg cm^−2^ in Cd up to 6.99 μg cm^−2^ in the control. Thus, the stress caused by heavy metals determines a significant reduction of chl b by 28–47% compared to the control plants, and Cd caused a 30–36% reduction in the values of this pigment compared to the rest of the variants.

The total chlorophyll amount recorded a variation between 18.22 μg cm^−2^ for Cd and 27.67 μg cm^−2^ in the control. In this case, too, there is a negative influence of the Cd treatment, which determined a significant reduction of the total quantity of chlorophylls by 34% compared to the control and by 26–29% compared to the other metal variants.

Under the heavy metals, carotenoids were significantly reduced by 18–24% compared to the recorded values for the control. Only for Pb pollution was there a significant increase of 15% compared to the control and by 34–52% compared to the other variants.

The total amount of photosynthetic pigments recorded values contained between 22.45 μg cm^−2^ at Cd and 32.58 μg cm^−2^ in the control, for Zn and Pb the variations were insignificant. In this case, Cd pollution also generated a significant reduction in the total amount of pigments.

Heavy metal pollution caused a significant increase in the amount of chl a compared to chl b (Figure 3). The ratio of chl a/b in the case of Pb pollution was significantly higher than in the plants of the other variants. In the case of unilateral pollution by Cu, Zn, or Cd, this ratio is close to level 4 specific to photosynthetic type I species. The analysis ratio between chlorophyll and carotenoids (chl a+b/car) shows that Cu and Zn pollution caused a significant increase in it, against the background of an increase in the number of chlorophyll pigments by 20–22% and a somewhat balanced reduction of carotenoids. Cd and Pb pollution caused an increase in carotenoids of 2.9–3.8% and a proportional reduction in chlorophylls compared to the control.

Given the relationships between correlations between photosynthetic pigments and metal concentration in leaves of S. perfoliatum plants (Table 3), it is noted that the Cd content was negatively and highly significantly correlated with both types of chlorophyll and total amounts of leaf pigments. In addition, the Zn content exhibited a strong negative correlation with car contents and chl b. This means that an increase in Cd in leaves was associated with a significant decrease in total chl content, and the presence of Zn has had a significant negative effect, particularly on the car content.

### 2.3. Evaluation of the Effects of Heavy Metal Exposure on Enzymatic Activity

Information from Appendix A confirms the homogeneity of the distribution of antioxidant enzymatic activities of superoxide dismutase (SOD), catalase (CAT), and peroxidase (POD) in different organs of S. perfoliatum plants under different pollutants. Analysis of variance performed for antioxidant enzymatic activities of SOD, CAT, and POD (Appendix A) revealed significant differences across plant organs and across treatments. However, the differences between plant organs regarding SOD under control were non-significant.

#### 2.3.1. SOD Activity

The SOD activity in plant roots was significantly reduced when exposed to different types of heavy metals in the nutrient solution. The values ranged from 136.93 U mg^−1^ protein min^−1^ for Cd and 202.07 U mg^−1^ protein min^−1^ for the control (Table 4). The stress generated by Cd and Pb contamination had a greater impact on SOD activity in roots compared to other treatments. For stems, only Cu, Zn, and Cd significantly reduced SOD activity by 22.6–26.8%, whereas Pb contamination results in similar activity levels to the control group, but significantly higher in comparison with other variants. In the leaf apparatus, in Pb-containing variants, SOD activity was significantly reduced from 207.48 U mg^−1^ protein min^−1^ in control plants to 169.46 U mg^−1^ protein min^−1^. Substrate contamination with Zn and Cd significantly enhanced SOD activity compared to Pb and Cu.

In the absence of heavy metals from the nutrient solution, the control plants showed a reduced and insignificant variation in SOD activity in the organs. Plants grown under Cu- or Zn-loaded substrate conditions showed significantly lower SOD activity in stems than in leaf apparatus and roots. On the other hand, exposure to Cd and Pb causes noticeable variations in SOD activity between the organs, with higher activity in leaves and lower activity in roots for 20 ppm Cd concentrations, and higher activity in stems and lower activity in roots for 400 ppm Pb.

#### 2.3.2. CAT Activity

When *S. perfoliatum* plants were exposed to heavy metals, increased CAT activity was observed in all three organs compared to control plants. Thus, in the root tissues, CAT activity showed a significant increase from 0.49 µmol H_2_O_2_ mg^−1^ protein min^−1^ in control plants to 3.18 µmol H_2_O_2_ mg^−1^ protein min^−1^ in the plants treated with Cd, with the effects of Zn and Pb being lower. The highest CAT activity was observed in the roots of plants grown on Zn and Cu substrates, showing a 45–170% increase compared to those grown on Cd and Pb. Similar results were observed in the leaf apparatus, where Zn and Cu caused a significant increase in CAT activity (19–98%) compared to the other two polluted variants.

CAT activity showed variations in distinct organs of control plants: 86% progressive increases between roots and stems and respectively, 98% between stems and leaves. Similar CAT activity was observed in plants treated with Cu and Zn, stronger in stems and weaker in roots. CAT activity in Cd-stressed plants was 30.5–35% lower in leaves than in stems and roots. In plants treated with Pb, CAT activity was 42–54% higher in roots than in stems and leaves.

#### 2.3.3. POD Activity

The results showed that under stress conditions induced by Cu, Zn, and Cd, POD activity in roots significantly increased by 17–71%, apart from Pb stress which caused a small and insignificant change of 7%. For variants treated with Cu, Zn, and Cd, POD activity in stems was 92–133% more intense than control or Pb. POD activity in the foliar apparatus of plants exposed to heavy metal stress was stronger, ranging from 245–257% for Zn and Cu.

Plants exposed to Cd stress recorded 176–191% higher POD antioxidant activity in roots and stems compared to leaves. Cu and Zn revealed notable differences in POD activity from one organ to another, higher in stems and lower in leaves. Pb excess was associated with significant intensification of POD activity in roots and 31–50% lower levels in stems and leaves. In the control, a major decrease in POD activity was observed in leaves compared to stems or roots.

#### 2.3.4. Correlations for the Activity of Enzyme Systems

Analysis of the correlations existing between the three determined enzymatic systems (Table 5) showed that the antioxidant activity of SOD was significantly negatively correlated with Cd, Pb, and total amounts of heavy metal in roots, as well as with Pb content and the total amount of heavy metals in leaves. In stems, a significant negative correlation between SOD activity and Zn content was reported. CAT activity in the roots was significantly positively correlated with Cd content, while in stems and leaves, it was positively associated with Zn content. Compared with SOD and CAT, the correlation of POD activity is more complex. Thus, the POD activity of the roots was highly positively associated only with Cd content, while in leaves it was significantly positively correlated with Cu, Zn, and total amounts of heavy metal. A similar positive correlation with POD activity in stems, and a significant negative correlation with Pb content, was observed.

Based on the variance components of multiple regressions from Appendix A, it was noted that a considerable part (94.05–94.91%) of the variation in SOD antioxidant enzyme activity in roots and stems could be explained by the effect of the four heavy metals included in the regression model. The Cd content in roots and stems had a significant effect (about 64%) on SOD activity. It was also noted that Pb content in roots and Cd content in stems had a significant effect on SOD activity. In leaves, due to a high contribution of Pb (59.5%), the combined effect of the four heavy metals could predict a significant variation in SOD activity (about 88%).

Regression analyses (Appendix A) indicated that 95.6–98.8% of the CAT activity variation was influenced by the four heavy metals. Therefore, Cd contributed 76% and played an important role in determining CAT activity in roots, while Zn content mainly determined 62–76.6% of variations for CAT activity in stems and leaves.

According to the variance components of the multiple regressions from Appendix A, the results indicated that the effect of heavy metals in the regression model could predict 96–98.7% of the variability in POD activity. Therefore, Cd has the highest contribution (42.9–65.9%) to POD activity in stems and roots, while the activity of this enzyme in leaves was mainly influenced by Cu.

The values of the Durbin Watson (DW) coefficient obtained from all regressions indicate that the errors related to the obtained results are not affected by the order of the four variables used in the regression equations. This means that the estimated results of the antioxidant activity of SOD, CAT, and POD remain unaffected.

In Figure 4, the biplot’s two PC axes explain 99.85% of the variation in the antioxidant activities of SOD, CAT, and POD in organs of *S. perfoliatum* plants under different treatments with heavy metals. The highest antioxidant activity of CAT and POD was found in the stems of plants treated with copper and zinc, while the lowest activity of SOD was observed in the same area. Furthermore, plants exposed to Cd stress also showed high levels of CAT and POD activity in their stems and roots. In contrast, under Cu and Zn stress, SOD activity was lowest in stems, whereas under Cd and Pb stress, SOD activity was reduced in roots. Except for plants treated with Pb, which exhibited the highest SOD activity in the stems, leaves had the highest SOD activity for all treatments.

## 3. Discussion

Heavy metals accumulated in the environment play a significant role in inducing stress in plants [51]. Plants have developed specific strategies and mechanisms over time to withstand stress caused by heavy metal pollution, such as: efflux pumps, cell sequestration, compartmentalization, binding of heavy metals in different structures, and/or production of strong ligands by phytochelation [52,53]. Numerous plant species are already known to be able to remediate soils contaminated with heavy metals. They are part of families such as: Brassicaceae [54,55,56], Asteraceae [57,58,59,60], Chenopodiaceae [61,62,63], and Scrophulariaceae [64,65].

Recent studies suggest that *Sperfoliatum* L., a plant with high ecological adaptability, rapid vegetative growth, and high values of bioaccumulation factors (BAF), translocation (TF), and removal efficiency (RE) [42], seems to fit into the specific pattern of hyperaccumulators [29,31,41,66].

Hydroponic systems represent an effective and modern technique for evaluating the phytoextractive potential of various plant species [67]. Using hydroponics offers several advantages including standardized and repeatable experimental conditions, high availability of heavy metals, high precision of analysis, and determination of the absorption, translocation, and accumulation of metals in plant organs and tissues [68,69,70].

### 3.1. Dynamics of Heavy Metal Accumulation in S. perfoliatum Plant Organs (Phytoextraction)

The amount of heavy metals accumulated in different plant species is determined by the ability to retain the compound or metal and mobilize between cells [71]. Accumulation represents a complex and dynamic process that involves several stages such as: transport of heavy metals through cellular plasma membranes at the level of the root system, passage into the xylem, translocation, isolation, and/or detoxification at the cellular level in the other organs [70].

The results obtained in this study show that the dynamics of accumulation in plants depend on the type of metal, the amount existing in the environment, and the organ. The control nutrient solution generated the accumulation of Zn mainly in the stems, Cu in the roots, and Pb and Cd in small amounts in the leaves. The pollution of the nutrient substrate with Cu, and respectively with Zn, determined the accumulation of larger amounts of these elements in the stems, more than twofold than in the roots and leaves.

In the case of the two studied micronutrients (Cu and Zn), different behaviors were observed. Copper accumulation was higher in the version polluted with this metal, while Zn absorption and accumulation were lower even though its concentration in the polluted substrate was three times higher (1200 ppm Zn/400 ppm Cu). In addition, the nutrient substrate polluted with Zn determined the absorption and accumulation of small amounts of Cu. These results suggest an antagonism between Cu and Zn from the moment of root absorption until passage through the cortex to the xylem. The Arabidopsis P-type ATPases HMA2/4 could be involved in this process because they serve as xylem-loading transporters for Cu and Zn. These mechanisms of antagonistic behavior and inhibition between Cu and Zn have also been observed in other studies [72,73,74].

Examining data on phytoextraction, it was noticed that the presence of Cd and Pb contamination led to the accumulation of these metals mainly in the roots. In the case of Pb, the accumulation values were particularly high. Moreover, when the substrate was contaminated with Cd, there was an increase in the accumulation of Cu and Pb ions in the roots. Similarly, Pb contamination caused the accumulation of Cu and Cd in the roots.

For Cd, the difference in accumulation between roots and stems is insignificant. As Cd pollution intensifies, the concentration of this metal in stems gradually increases and the concentration differences between stems and roots decrease, according to previous studies [75,76]. The main mechanisms of plant tolerance to Cd stress include the complexation of this metal with sulfur-rich peptides and organic acids, as well as its storage in the vacuolar system [77]. Studies have shown that more than half of Cd accumulates in the cell wall, which contains protein compounds and polysaccharides. Negatively charged sites on the surface of the wall bind to Cd, limiting its mobility in plants [78].

In the case of Pb pollution, the plants accumulate it mainly in roots, and in the aerial organs, especially in the leaf apparatus. *S. perfoliatum* has also confirmed the tendency to absorb significant amounts of Pb from the polluted substrate and accumulates it in the roots [78,79,80].

The combined and simultaneous action of four heavy metals had a detrimental impact on *S. perfoliatum* plants. Therefore, within 48 h, plants registered obvious toxicity phenomena manifested by depigmentation (chlorosis and necrosis), the withering of leaves, the appearance of necrotic spots on the stems, reduction of tissue turgor, and loss of their erect position. Thus, the plants were harvested after 48 h to evaluate the accumulation of each metal in the organs. Under the effect of the combined action of Cu, Zn, Cd, and Pb, the accumulation of these elements in different organs depends on the metal type and its exogenous concentration. The order of accumulation in roots was Pb > Cu > Cd > Zn, in stems Cu > Cd > Zn and >Pb, and in leaves Cu > Pb > Zn > Cd. These results attest to the complex interaction between the four metals and their synergistic polluting effect, altering the accumulation dynamics as compared to the unilateral action of each element. The comparable behavior of Cd and Zn may be due to the similar physical and chemical properties of the two heavy metals as well as their atomic mass. Moreover, Zn can diminish the harmful effects of Cd by promoting its absorption and accumulation in the roots [81,82].

### 3.2. Evaluation of the Impact of Heavy Metal Accumulation on Free Proline and Assimilatory Pigments Content

Proline plays an important role in increasing plant stress tolerance and may be useful in reducing the adverse effects caused by various heavy metal toxicities [83]. Due to its metal chelating properties, proline acts as a molecular chaperone, an antioxidant defense molecule that scavenges reactive oxygen species (ROS), as well as having a signaling behavior to activate specific gene functions that are crucial for plant recovery from stress [84,85]. It also acts as an osmoprotectant, a potential source for nitrogen and carbon acquisition, and plays a significant role in plant development [86]. The increase in the amount of proline was recorded in all variants, with the highest value observed in their combined applications. However, in the case of Zn and Cu pollution, the increase in free proline levels was insignificant compared to the control.

Despite previous research [87,88] indicating that plant species tolerant to metal contamination exhibit higher proline accumulation in comparison to sensitive ones, the proline content of *S. perfoliatum* did not significantly increase under Cu and Zn pollution (Figure 2). This suggests that the high tolerance of *S. perfoliatum* to Cu and Zn pollution, proved by previous research [41,42], cannot be solely attributed to the amount of accumulated proline.

In the case of substrates contaminated with Cd and Pb, as well as under the action of all four metals combined, a significant accumulation of free proline in various plant organs was observed. The results obtained are consistent with other studies [89,90] which demonstrated that proline accumulation in Cd- and Pb-stressed plant tissues could be responsible for counteracting Pb-mediated lipid peroxidation and membrane alterations [91]. Proline accumulates in plants in response to metal stress, and performs multiple roles such as osmotic protector, molecular chaperone, and antioxidant, and provides the maintenance of the redox balance by contributing to the reduction of free radicals [92,93,94].

The total amount of chlorophyll pigments was not significantly affected by heavy metal presence in the cultivation substrate, except Cd. Chlorophyll b seems to be more sensitive to the action of pollutants, especially Cd, where the amount was halved compared to the control (6.88–3.64 µg cm^−2^). The decrease in the content of chlorophyll pigments due to heavy metal stress may be attributed to the inhibition of enzymes that are responsible for their biosynthesis. Previous studies have indicated that Cd affects the biosynthesis of chlorophylls and inhibits the synthesis of protochlorophyll reductase and aminolevulinic acid.

The ratio chl a/chl b, often used as an indicator of stress, was increased in all metal-containing variants. This has also been confirmed by previous studies on heavy metals in *E. nigrum* (Ericaceae) leaves [95], *S. oleracea* (Chenopodiaceae) [96], *P. vulgaris* (Fabaceae) [97], etc. High values of the chl a/chl b ratio indicate a change in the PSII/PSI ratio in stressed leaves [98]. Therefore, under stress conditions caused by the accumulation of heavy metals in plant leaf apparatus, chlorophyll pigments undergo a series of photochemical reactions, such as oxidation, reduction, and pheophytinization [99].

Additionally, the ratio of chlorophyll pigments to carotenoids increased compared to the control in the case of Cu and Zn contamination and decreased in the case of Cd and Pb, indicating a higher amount of carotenoids and therefore a higher need for photoprotection by this non-enzymatic antioxidant. According to previous studies, one of the essential functions of carotenoids is to protect plant cells. Among the antioxidant molecules of the plant defense system against oxidative damage, carotenoids are very efficient in scavenging ROS, having an important role in concentrating the harvested light energy to the plant photosystem and in using non-phytochemical quenching to dissipate excess light. Apart from their structural functions in photosynthetic reaction centers and antennae, carotenoids play an essential role in protecting the photosynthetic system from oxidative damage by scavenging reactive oxygen species [100,101].

### 3.3. Evaluation the Impact of Heavy Metal Accumulation on Enzymatic Activity

Plants have many antioxidant molecules and enzymes that help protect them from oxidative damage and control the levels and effects of Reactive Oxygen Species (ROS). They have the ability to regenerate the active form of antioxidants and eliminate or reduce the damage caused by ROS. Protection against oxidative stress is achieved through the production of enzymatic antioxidants such as superoxide dismutase (SOD), peroxidase (POD), and catalase (CAT), while glutathione, carotenoids, and ascorbate represent non-enzymatic components [102,103] cited by Bhaduri and Fulekar, 2008 [104].

The activation of antioxidant enzymes like superoxide dismutase (SOD) and catalase (CAT) are some of the major mechanisms of metal detoxification in plants. The results obtained showed a decrease in SOD activity and an increase in CAT activity in all contaminated variants and in all plant organs. The reduction in SOD activity was most significant in roots exposed to Cd, stems exposed to Zn and Cd, and leaves exposed to Pb. Meanwhile, the accumulation of heavy metals, especially Cd in roots and Zn in stems and leaves, intensifies catalase activity. These findings are consistent with previous research conducted on species within the Brassicaea family [105,106,107]. Furthermore, metal accumulation also intensifies POD activity in different plant organs, reaching maximum levels for Cd in roots and Cu in stems and leaves. The lowest values of metal accumulation were observed in organs exposed to Pb.

When enzyme activity decreases as a result of prolonged exposure or higher concentrations, it suggests that the plant’s antioxidant defense capacity has been surpassed by toxic effects. Although the increase in antioxidant defense enzyme activity can be attributed to their induction by heavy metal toxicity, the decrease in their activity can also be explained by the disruption of anti-oxidative mechanisms due to toxicity. The quantification and determination of threshold levels for both effects are complicated due to their dependence on numerous factors [108].

Heavy metals induce an imbalance in plants causing oxidative stress by generating harmful oxygen radicals that cannot be removed by the antioxidant defense mechanism. This mechanism adapts by upregulating anti-oxidative enzymes such as POD, CAT, and SOD, which provide an effective detoxification and scavenging system for toxic oxygen species [109].

The research results show that *S. perfoliatum* plants tried to protect against heavy metal exposure by activating the antioxidant enzyme system, and the level of activity seems to be directly dependent on the type of heavy metal, concentration, and exposure time.

## 4. Materials and Methods

### 4.1. Biological Material

One-year-old plants of *S. perfoliatum*, in the stage of vegetative mass growth (35–37 BBCH), were used in this study. These plants were obtained from seeds and were part of the collection of the Plant Physiology department.- University of Life Sciences “King Mihai I” from Timișoara. *S. perfoliatum* is recognized to be a suitable option for the restoration of soils contaminated with heavy metals because of its fast growth rate, high biomass production, ecological adaptability, and tolerance to stress, etc. [30,31,40,41,110].

### 4.2. Floating Hydroponic System

For the experiment, 25 plants with uniform growth and development were chosen. The plants were washed to remove soil and then placed in Plexiglas parallelepiped pots measuring 150/150/300 mm. Each pot was filled with 5 L of solution, and only one plant was placed in each pot. Oxygen for root respiration was provided through a bubbler insufflating air with a capacity of 20 cm^3^ vat^−1^ min^−1^. The plants were fixed using floating polystyrene plates on the surface of the liquid static medium. The pots were covered with aluminum foil to prevent overheating of the nutrient solutions and to keep the roots in the dark.

The experiment was conducted in greenhouse conditions for 20 days in April 2022, with a completely randomized design (temperature 25 ± 5/18 ± 5 °C day/night, relative humidity 65 ± 5%, and photoperiod of approximately 14 h). The experimental variants were Control—Hoagland nutrient solution without heavy metals; Hoagland solution with 400 ppm Cu; with 20 ppm Cd; with 400 ppm Pb; with 1.200 ppm Zn, and Hoagland solution with all four heavy metals added in specific individual concentrations. Each experimental variant had five replicates. Concomitant addition of metals to the solution caused plant death after only 48 h.

### 4.3. Preparation of Nutrient Solutions

Numerous studies have focused on the harmful effects of heavy metals on plants grown in contaminated soil substrates. However, cultivation techniques in hydroponic systems are becoming increasingly important due to their effective bioavailability of metals compared to solid substrates [111].

The selection of extraction solutions depends on various factors such as the soil’s physical and chemical properties, the element chemistry, and the response of different plant species [112]. Over the last few decades, comparative studies have been performed to prove that floating hydroponic systems and soil are interchangeable techniques, and the results have demonstrated that metal availability is much greater in hydroponics [113,114].

Reagents; ammonium dihydrogen phosphate (NH_4_H_2_PO_4_), potassium nitrate (KNO_3_), calcium nitrate tetrahydrate (Ca(NO_3_)_2_·4H_2_O), magnesium sulfate heptahydrate (MgSO_4_·7H_2_O), boric acid (H_3_BO_3_), manganese (II) chloride dihydrate (MnCl_2_·2H_2_O), zinc sulfate heptahydrate (ZnSO_4_·7H_2_O), copper (II) sulfate pentahydrate (CuSO_4_·5H_2_O), molybdic acid (H_2_MoO_4_·H_2_O assaying 85% MoO_3_), EDTA—ethylenediaminetetraacetic acid ([CH_2_N(CH_2_CO_2_H)_2_]_2_), potassium hydroxide (KOH), iron (II) sulfate heptahydrate (FeSO_4_·7H_2_O), cadmium acetate dihydrate (CH_3_COO)_2_Cd·2H_2_O), lead (II) acetate trihydrate (Pb(CH_3_COO)_2_·3H_2_O).

The basic solutions with macro (N, P, K) and secondary elements (Ca, Mg, S), dissolved in 1 L of distilled water; NH_4_H_2_PO_4_ 1M—115.03 g KNO_3_ 1M—101.10 g, Ca (NO_3_)_2_·4H_2_O 1M—236.15 g, MgSO_4_·7H_2_O 1M—246.47 g.

Stock solution of microelements (salts dissolved in 1-L distilled water); 2.86 g H_3_BO_3_, 1.81 g MnCl_2_·2H_2_O, 0.22 g ZnSO_4_·7H_2_O, 0.08g CuSO_4_·5H_2_O, 0.02 g H_2_MoO_4_ H_2_O. For preparing the iron stock solution, we followed the steps:−*solution 1*: 26.1 g EDTA was dissolved in 286 mL distilled water containing ~19g KOH.−*solution 2*: 24.9 g FeSO_4_·7H_2_O was dissolved in 500 mL distilled water. Then, solution 2 was added slowly to solution 1 while shaking it. The mixture was left to air overnight while being shaken. This resulted in a increase in pH to 7.1 and a red color with slight precipitation. Finally, the solution was brought to a volume of 1 L and stored in a dark place.

Stock solutions of polluting metal ions (salts dissolved in 1-L distilled water); Cu—16 g/L Cu^2+^—62.87g CuSO_4_·5H_2_O, Cd—2 g/L Cd^2+^ (4.74 g CH_3_COO)_2_Cd·2H_2_O, Pb—16 g/L Pb^2+^ 29.27 g Pb (CH_3_COO)_2_·3H_2_O, Zn—48 g/L Zn^2+^ 211.15 g ZnSO_4_·7H_2_O (Table 6).

### 4.4. Plant Analysis

The process of collecting plants for analysis was divided into two stages. In the first stage, after 48 h, the plants were exposed to the combined stress of four pollutants. At this stage only the level of metal accumulation in the organs was measured, because the plants were no longer viable. The second stage of harvesting was done after 20 days, for the other experimental variants, including the control and the individual and combined effects of Cu, Cd, Zn, and Pb metals. Analysis was carried out for each variant in five replicates.

#### 4.4.1. Determination of the Heavy Metal’s Accumulation in Plant Organs

Dried plant organs (roots, stems, and leaves) were calcined to remove their organic parts in order to determine heavy metal accumulations using the method described by Sumalan et al. [41]. Heavy metal content was determined by the AAS method (Varian SpectrAA 280FS, Agilent Technologies Inc. Santa Clara, CA, USA) according to ISO 11047/1998 (https://www.iso.org/obp/ui/#iso:std:iso:11047:ed-1:v1:en accessed on 17 May 2019) [115].

#### 4.4.2. Determination of Free Proline and Assimilatory Pigments Contents

Determination of free proline was performed on roots, stems, and leaves according to the method of Bates et al. [116]. Briefly, a homogenate of the resulting sample was prepared with 3% sulfosalicylic acid and colored pink by reaction with glacial acetic acid and ninhydrin. Color intensity was determined spectrophotometrically at 546 nm.

Determination of chlorophylls and carotenoid pigments in the foliar apparatus was performed by extraction in 96% ethanol (*v*/*v*) with a solvent/biomass ratio of 1:10 (*w*/*v*). The absorbance was read at 664 nm (chlorophyll a), 649 nm (chlorophyll b), and 470 nm (carotenoids). The pigment content was calculated according to Lichtenthaler et al. [117].

#### 4.4.3. Determination of Enzyme Activities

Enzyme extracts for SOD, CAT, and POD activities were prepared by freezing 1 g of leaf samples in liquid nitrogen, followed by grinding with 5 mL extraction buffer (0.1 M phosphate buffer, pH 7.5, containing 0.5 mM EDTA). A total of 1 g of leaf samples were frozen in liquid nitrogen followed by grinding with 10 mL extraction buffer (0.1 M phosphate buffer, pH 7.5, containing 0.5 mM EDTA). The homogenates were filtered through four layers of cheesecloth and then centrifuged at 4 °C for 20 min at 15,000× *g*. The supernatants were collected and used to assay enzyme activities. All steps in the preparation of enzyme extracts were performed at 4 °C [118].

Superoxide dismutase (SOD) (EC 1.15.1.1) activity was determined through the photochemical NBT method, which measures the ability of this enzyme to prevent the reduction of nitro blue tetrazolium (NBT) [119]. The reaction was prepared in 1 mL final volume, containing 50 mM sodium phosphate buffer pH 7.8, 13 mM methionine, 0.1 mM EDTA, 75 µM NBT, 2 µM riboflavin, and enzyme extract. To avoid degradation, riboflavin was added last. The reaction was started by placing the samples under a light source (15 W fluorescent lamp) for 15 min. The following two blanks were prepared: one without enzyme extract placed under the light in order to completely develop the reaction, and another one containing the enzyme extract placed in the dark to avoid the reaction. The absorbance was recorded at 560 nm, and the enzyme activity was expressed in unit mg^−1^ protein min^−1^.

Catalase (CAT) (EC 1.11.1.6) activity. The samples were collected from different plant organs and crushed in a mortar pestle using 100 mM phosphate buffer (5 mL, pH 7.5). The resulting extract was centrifuged for 10 min at 4 °C and the obtained supernatant was used for the enzyme assay. CAT activity was measured spectrophotometrically at an absorbance rate of 230 nm for 3 min and expressed in µmol H_2_O_2_ mg^−1^ protein min^−1^ [120]. The reaction mixture contains enzyme extract (100 µL), 50 mM phosphate buffer (pH 7.00), and H_2_O_2_ (0.1 mL, 10 mM).

Peroxidase (POD) (EC 1.11.1.7) activity was determined spectrophotometrically at 470 nm, indicating the formation of tetra-guaiacol from guaiacol in the presence of H_2_O_2_. The enzyme activity was calculated as per extinction coefficient of its oxidation product, tetra-guaiacol (ε = 26.6 mM^−1^ cm^−1^). The reaction mixture consists of 83. 50 mM phosphate buffer (pH 6.1), 16 mM guaiacol, and 0.1 mL enzyme extract, in which 2 mM H_2_O_2_ is added to initiate the reaction. The mixture was diluted with distilled water to a final volume of 3.0 mL. Enzyme-specific activity was expressed as µmol guaiacol mg^−1^ protein min^−1^.

### 4.5. Data Analyses

Data representing mean ± standard error (SE) were calculated based on three replicate samples. Data from all analyses and determinations were statistically processed using ANOVA, and mean values were compared using the Tukey HSD (Honestly Significant Difference) test [121]. The significance of the differences and the difference between means considered significant (*p* < 0.05) were marked by different letters. The homogeneity of variances was analyzed using Cochran’s C and Bartlett’s tests.

Relationships between photosynthetic pigments and metal concentration in leaves or between enzymatic activities and metal concentration in different organs were analyzed using Pearson’s correlation coefficient and multiple regressions. The significance of correlation coefficients was analyzed by a two-tailed test. Principal component analysis (PCA) for enzymatic activities under different heavy metal treatments was performed using MATMODEL Version 3.

## 5. Conclusions

Phytoremediation is a promising technique for cleaning up heavy metal-contaminated ecosystems and offers several benefits over traditional methods. *S. perfoliatum* L. is a species that shows potential for remediating polluted environments due to its fast growth rate, high biomass production, ecological flexibility, stress tolerance, and usefulness in biogas production. Growing plants in hydroponic systems with heavy metal additives has advantages such as providing plants with adequate mineral elements (including heavy metals), standardizing experimental conditions, and accurately determining the absorption and accumulation of heavy metals in plant tissues.

The dynamics of the accumulation in plants of Cu, Zn, Cd, and Pb depend on the type of metal, the amount present in the environment, and the vegetative organ. The obtained results suggest the existence of an absorption antagonism between Cu and Zn and an accumulation of Cd and Pb, especially at the level of the roots.

Free proline accumulation is more intense when the nutrient solution is polluted with Cd and Pb, but is reduced in the case of Cu and Zn pollution, with differentiations depending on the organ. Although the total amount of chlorophyll pigments is not significantly affected by the accumulation of pollutants in the leaf apparatus, this fact determined the change in the ratio between chlorophyll a and b in favor of the former, and increased the amount of carotenoids through the accumulation of Cd and Pb in the organs. These results confirm the importance of carotenoids in the plant’s defense mechanism against stress caused by the accumulation of heavy metals.

The evaluation of the enzyme activity demonstrated a different behavior of the studied enzymes, some being stimulated and others being inhibited by the accumulation of heavy metals in the tissues. The intensity of the enzyme system’s activity seems to be directly dependent on the plant species, organ, metal type, concentration, and time of exposure to stress. Therefore, for a comprehensive understanding of these aspects, extensive research on the physiological mechanisms involved in the phytoremediation of ecosystems polluted with heavy metals is needed.

## Figures and Tables

**Figure 1 plants-12-01718-f001:**
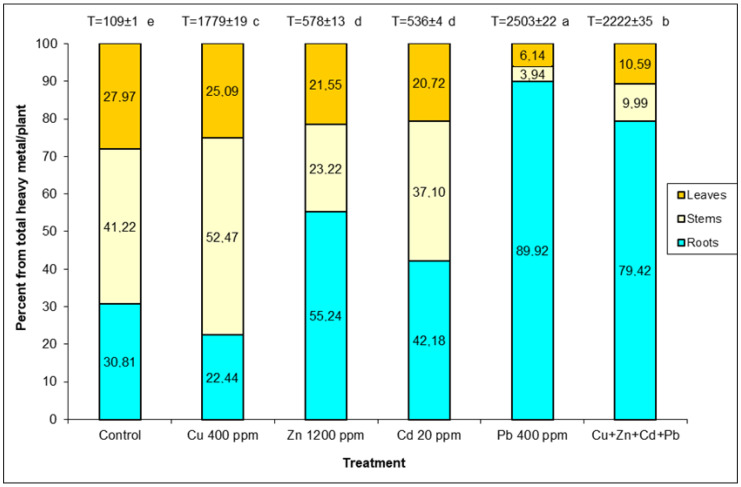
Percentage distribution of accumulated heavy metals in organs of *S. perfoliatum* plants grown under different treatments. T-total accumulated heavy metals (μg g^−1^ DW) in plant. Means (±SE) with different letters are significant at *p* < 0.05 according to Tukey test.

**Figure 2 plants-12-01718-f002:**
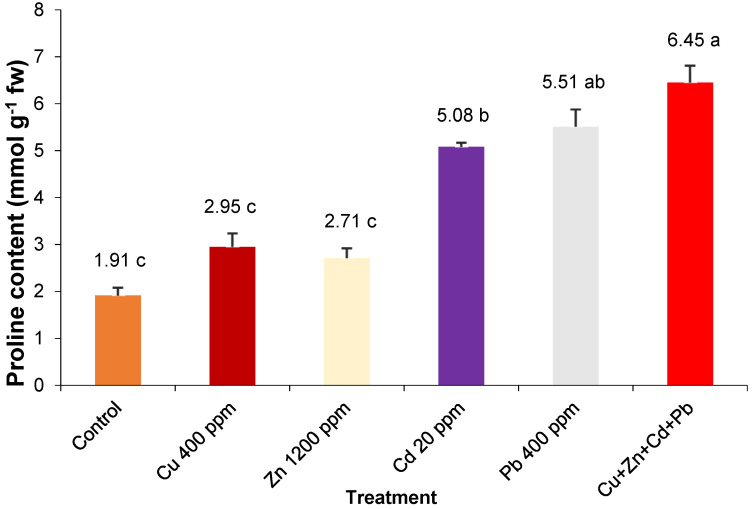
Proline content of entire *S. perfoliatum* plants grown under different treatments. Error bars represent SE. Means with different letters are significant at *p* < 0.05. HSD5% = 1.28.

**Figure 3 plants-12-01718-f003:**
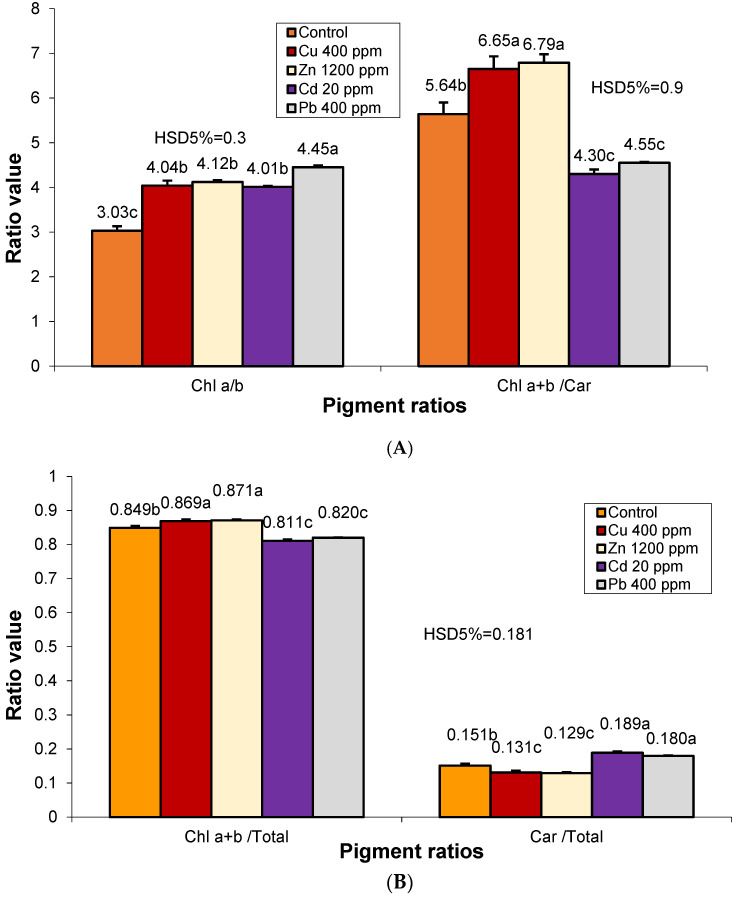
Changes in photosynthetic pigment ratios in *S. perfoliatum* leaves under different treatments. Data represent mean ± SE. Means with different letters are significant at *p* < 0.05 according to Tukey test: (**A**) Chl a/b and Chl a+b/Car; (**B**) Chl a+b/T and Car/T. Chl-Chlorophyll; Car-Carotenoids; T-Total amount of photosynthetic pigments.

**Figure 4 plants-12-01718-f004:**
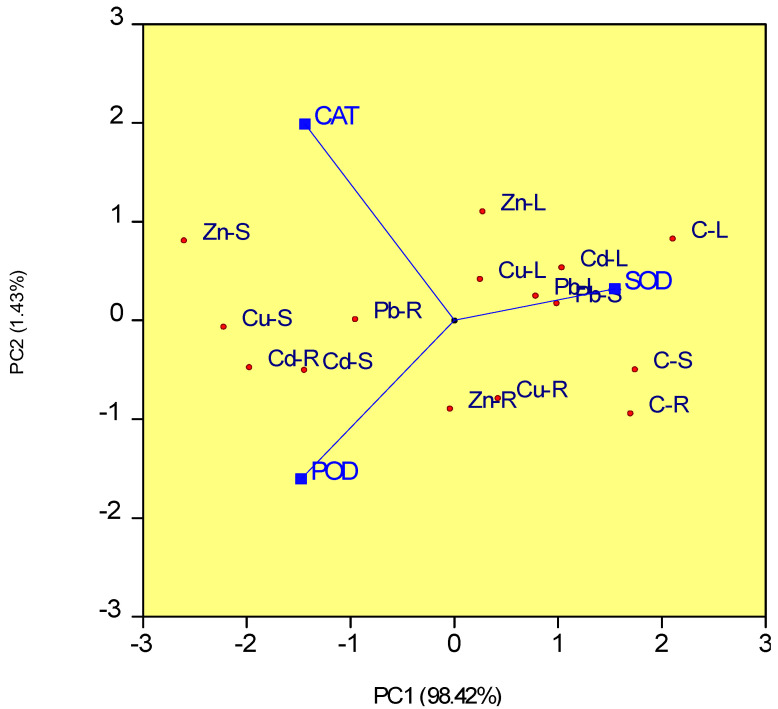
Principal component analysis (PCA) for antioxidant enzymatic activities of SOD, CAT) and POD in different organs of *S. perfoliatum* plants under control and heavy metals. C-R, roots of control plants; C-S, stems of control plants; C-L, leaves of control plants; Cu-R, roots of plants treated with Cu 400 ppm; Cu-S, stems of plants treated with Cu 400 ppm; Cu-L, leaves of plants treated with Cu 400 ppm; Zn-R, roots of plants treated with Zn 1200 ppm; Zn-S, stems of plants treated with Zn 1200 ppm; Zn-L, leaves of plants treated with Zn 1200 ppm; Cd-R, roots of plants treated with Cd 20 ppm; Cd-S, stems of plants treated with Cd 20 ppm; Cd-L, leaves of plants treated with Cd 20 ppm; Pb-R, roots of plants treated with Pb 400 ppm; Pb-S, stems of plants treated with Pb 400 ppm; Pb-L, leaves of plants treated with Pb 400 ppm.

**Table 1 plants-12-01718-t001:** Heavy metal concentrations in organs of *S. perfoliatum* plants grown under different treatments.

Treatment		Plant		Metal	
	Organ	Cu	Zn	Cd	Pb
Control	Roots	8.16 ± 0.30 a	23.50 ± 1.18 b	0.56 ± 0.04 b	1.51 ± 0.08 a
	Stems	4.81 ± 0.28 b	39.83 ± 1.73 a	0.18 ± 0.01 c	0.30 ± 0.03 c
	Leaves	3.63 ± 0.21 c	24.44 ± 1.01 b	2.03 ± 0.08 a	0.52 ± 0.03 b
	*HSD5%*	*1.16*	*5.83*	*0.22*	*0.20*
Cu	Roots	333.81 ± 7.41 b	39.84 ± 1.42 c	3.77 ± 0.17 b	21.77 ± 0.75 a
400 ppm	Stems	810.78 ± 10.15 a	108.79 ± 4.36 a	9.84 ± 0.27 a	4.02 ± 0.13 c
	Leaves	360.94 ± 9.45 b	73.43 ± 2.04 b	2.14 ± 0.08 c	9.73 ± 0.37 b
	*HSD5%*	*39.41*	*4.06*	*0.83*	*2.13*
Zn	Roots	18.18 ± 0.73 a	73.43 ± 1.64 b	8.86 ± 0.38 a	218.56 ± 5.54 a
1200 ppm	Stems	1.93 ± 0.11 c	120.60 ± 5.36 a	5.44 ± 0.18 b	6.13 ± 0.15 c
	Leaves	4.07 ± 0.11 b	75.78 ± 1.22 b	3.18 ± 0.14 c	41.42 ± 0.95 b
	*HSD5%*	*1.87*	*14.38*	*0.88*	*14.08*
Cd	Roots	33.86 ± 1.05 a	28.95 ± 1.10 c	159.89 ± 4.42 a	3.60 ± 0.14 a
20 ppm	Stems	6.47 ± 0.25 c	36.87 ± 1.82 b	155.18 ± 3.65 a	0.50 ± 0.04 b
	Leaves	15.42 ± 0.39 b	55.45 ± 1.39 a	37.07 ± 1.61 b	3.20 ± 0.14 a
	*HSD5%*	*2.87*	*6.37*	*14.92*	*0.50*
Pb	Roots	159.01 ± 3.27 a	66.03 ± 1.61 a	1.48 ± 0.05 a	2024.24 ± 17.52 a
400 ppm	Stems	9.64 ± 0.42 c	55.73 ± 1.48 b	0.79 ± 0.04 b	32.48 ± 1.44 b
	Leaves	43.56 ± 0.84 b	44.80 ± 1.01 c	0.42 ± 0.02 c	64.82 ± 2.04 b
	*HSD5%*	*8.51*	*6.03*	*0.16*	*44.32*
Cu+Zn+	Roots	296.53 ± 7.67 b	75.66 ± 1.69 b	122.97 ± 3.48 b	1279.58 ± 22.85 a
Cd+Pb	Stems	402.18 ± 8.56 a	114.15 ± 2.82 a	143.84 ± 4.29 a	39.83 ± 1.61 b
	Leaves	81.52 ± 1.44 c	70.05 ± 2.15 b	7.08 ± 0.09 c	71.42 ± 2.30 b
	*HSD5%*	*29.02*	*9.84*	*13.85*	*57.68*

Data (μg g^−1^) represents mean ± SE. Means with different letters (a, b, c) are significant at *p* < 0.05 according to Tukey test.

**Table 2 plants-12-01718-t002:** Photosynthetic pigments in *S. perfoliatum* leaves under different treatments.

Treatment	Pigment				
	Chl a	Chl b	Chl a+b	Car	Total
Control	20.79 ± 0.88 a	6.88 ± 0.32 a	27.67 ± 1.13 a	4.91 ± 0.13 b	32.58 ± 1.16 a
Cu 400 ppm	19.80 ± 0.46 a	4.90 ± 0.08 b	24.69 ± 0.48 a	3.73 ± 0.16 c	28.42 ± 0.56 b
Zn 1200 ppm	20.45 ± 0.39 a	4.97 ± 0.14 b	25.42 ± 0.52 a	3.75 ± 0.07 c	29.16 ± 0.53 ab
Cd 20 ppm	14.58 ± 0.54 b	3.64 ± 0.13 c	18.22 ± 0.66 b	4.23 ± 0.06 c	22.45 ± 0.72 c
Pb 400 ppm	21.06 ± 0.74 a	4.74 ± 0.21 b	25.80 ± 0.95 a	5.67 ± 0.23 a	31.47 ± 1.17 ab
*HSD5%*	*2.92*	*0.90*	*3.67*	*0.67*	*4.08*

Data (μg cm^−2^) represent mean ± SE. Means with different letters are significant at *p* < 0.05 according to Tukey test. Chl-Chlorophyll; Car-Carotenoids.

**Table 3 plants-12-01718-t003:** Pearson correlations between photosynthetic pigments and metal concentration in leaves of *S. perfoliatum* plants.

Photosynthetic Pigments	Metal				
	Cu	Zn	Cd	Pb	Total
Chl a	0.101	−0.130	−0.917 ***	0.449	0.069
Chl b	−0.100	−0.549 *	−0.627 *	−0.194	−0.267
Chl a+b	0.044	−0.272	−0.890 ***	0.277	−0.033
Car	−0.409	−0.729 **	−0.193	0.422	−0.440
Total	−0.044	−0.399	−0.854 ***	0.340	−0.120

* Significant at *p* ≤ 0.05; ** Significant at *p* ≤ 0.01; *** Significant at *p* ≤ 0.001; *n* = 15. Chl-Chlorophyll; Car-Carotenoids.

**Table 4 plants-12-01718-t004:** Effect of heavy metal treatments on enzymatic activities of superoxide dismutase (SOD), catalase (CAT), and peroxidase (POD) in different organs of S. perfoliatum plants.

Antioxidant Enzymatic Activities	Treatment	Plant Organ			*HSD5%*
		Roots	Stems	Leaves	
SOD	Control	202.07 ± 4.92 a ^x^	202.07 ± 4.92 a ^x^	207.48 ± 3.88 a ^x^	*20.78*
	Cu 400 ppm	173.16 ± 2.62 b ^x^	156.39 ± 3.10 b ^y^	175.11 ± 2.38 cd ^x^	*11.78*
	Zn1200 ppm	180.41 ± 3.15 b ^x^	147.86 ± 4.23 b ^y^	190.12 ± 3.80 b ^x^	*16.30*
	Cd 20 ppm	136.93 ± 2.43 c ^z^	149.38 ± 3.36 b ^y^	188.02 ± 2.14 bc ^x^	*11.69*
	Pb 400 ppm	140.11 ± 4.15 c ^z^	187.25 ± 3.61 a ^x^	169.46 ± 2.39 d ^y^	*15.04*
	*HSD5%*	*16.67*	*18.78*	*14.04*	
CAT	Control	0.49 ± 0.02 e ^z^	0.91 ± 0.03 e ^y^	1.81 ± 0.05 d ^x^	*0.16*
	Cu 400 ppm	1.32 ± 0.04 d ^z^	4.19 ± 0.11 b ^x^	2.63 ± 0.12 b ^y^	*0.41*
	Zn1200 ppm	1.79 ± 0.05 c ^z^	5.22 ± 0.14 a ^x^	3.55 ± 0.09 a ^y^	*0.44*
	Cd 20 ppm	3.18 ± 0.07 a ^x^	2.89 ± 0.08 c ^x^	2.21 ± 0.06 c ^y^	*0.31*
	Pb 400 ppm	2.75 ± 0.08 b ^x^	1.93 ± 0.09 d ^y^	1.79 ± 0.05 d ^y^	0.33
	*HSD5%*	*0.26*	*0.45*	*0.37*	
POD	Control	7.76 ± 0.14 d ^x^	6.58 ± 0.18 c ^y^	1.81 ± 0.07 c ^z^	*0.59*
	Cu 400 ppm	9.08 ± 0.18 c ^y^	15.34 ± 0.37 a ^x^	6.46 ± 0.21 a ^z^	*1.15*
	Zn1200 ppm	11.84 ± 0.33 b ^y^	13.43 ± 0.38 b ^x^	6.24 ± 0.23 a ^z^	*1.38*
	Cd 20 ppm	13.29 ± 0.40 a ^x^	12.63 ± 0.37 b ^x^	4.57 ± 0.18 b ^y^	1.44
	Pb 400 ppm	8.32 ± 0.21 cd ^x^	5.77 ± 0.16 c ^y^	4.18 ± 0.13 b ^z^	*0.73*
	*HSD5%*	*1.26*	*1.43*	*0.80*	

Data (SOD-U mg^−1^ protein min^−1^; CAT- µmol H_2_O_2_ mg^−1^ protein min^−1^; POD- µmol guaiacol mg^−1^ protein min^−1^) represent mean ± SE. Different letters (a, b, c, d, e) in the column indicate significant differences (*p* < 0.05) between treatments according to Tukey test. Different superscript letters (x, y, z) in the row indicate significant differences (*p* < 0.05) between plant organs according to Tukey test.

**Table 5 plants-12-01718-t005:** Pearson correlations between antioxidant enzyme activities of superoxide dismutase (SOD), catalase (CAT), and peroxidase (POD) and metal concentration in different organs of *S. perfoliatum* plants.

Antioxidant Enzymatic Activity	Plant Organ	Metal				
		Cu	Zn	Cd	Pb	Total
SOD	Roots	−0.155	−0.182	−0.579 *	−0.525 *	−0.564 *
	Stems	−0.268	−0.544 *	−0.464	0.291	−0.396
	Leaves	−0.468	−0.454	0.105	−0.570 *	−0.606 *
CAT	Roots	−0.053	0.281	0.660 **	0.436	0.474
	Stems	0.372	0.868 ***	0.003	−0.196	0.450
	Leaves	0.125	0.831 ***	−0.095	0.114	0.244
POD	Roots	−0.356	0.080	0.777 ***	−0.367	−0.349
	Stems	0.590 *	0.657 **	0.296	−0.542 *	0.682 **
	Leaves	0.532 *	0.964 ***	0.004	0.244	0.690 **

* Significant at *p* ≤ 0.05; ** Significant at *p* ≤ 0.01; *** Significant at *p* ≤ 0.001; n = 15.

**Table 6 plants-12-01718-t006:** Quantities required for the preparation of 20 L of Hoagland nutrient solution.

Nutrient Solution Type	NH_4_H_2_PO_4_ 1M mL/20 L	KNO_3_ 1M mL/20 L	Ca (NO_3_)_2_·4H_2_O 1M mL/20 L	MgSO_4_·7H_2_O 1M mL/20 L	Microelements Stock Solution mL/20 L	Iron Stock Solution mL/20 L	16 g/L Cu^2+^ mL/20 L	2 g/L Cd^2+^ mL/20 L	16 g/L Pb^2+^ mL/20 L	48 g/L Zn^2+^ mL/20 L
Control	20	120	80	40	20	5	-	-	-	-
400 ppm Cu^2+^	20	120	80	40	20	5	25	-	-	-
20 ppm Cd^2+^	20	120	80	40	20	5	-	10	-	-
400 ppm Pb^2+^	20	120	80	40	20	5	-	-	25	-
1200 ppm Zn^2+^	20	120	80	40	20	5	-	-	-	25
Cu+Cd+Pb+Zn	20	120	80	40	20	5	25	10	25	25

## Data Availability

Not applicable.

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
