# Peer review of "The Impact of Heavy Metal Accumulation on Some Physiological Parameters in Silphium perfoliatum L. Plants Grown in Hydroponic Systems"

_plants, 2023, doi:10.3390/plants12081718_

Round 1
Reviewer 1 Report
Minor comments:
The article could benefit from editing regarding English language usage and efficient scientific writing. For example, use italics for scientific species names, the botanic family name isn’t in italics, etc.
Abstract
lines 24 and 25 modify as follows “with addition of (ppm) Cu-400, Zn -1200, Cd-20, 24 Pb-400, Cu+Zn+Cd+Pb (400/1200/20/400)
lines 30-32 modify as follows “For Cu and Zn, the accumulation (mean ± S.E. ppm) occurs mainly in the stems (810.78 ± 10.15 and 120.60 ± 5.36), for Cd in the roots and stems (159.89 ± 4.42 and 155.18 ± 3.65), while Pb accumulates mainly in roots (2024.24 ± 17.52).
Introduction
I think the introduction section should be shortened and revised better.
line 77, exchange “sol” for soil
line 86, exchange “Pe” for Per
line 98, delete comma in etc.
line 165 exchange …(cup plants) for (Asteraceae)
line 173 exchange Silphium perfoliatum (Asteraceae) for S. perfoliatum …..
line 176 exchange (Cu; Zn; Cd and Pb) for (Cu, Zn, Cd and Pb)
Results
The results section includes some points of the introduction, it should only include results. Also, include the ANOVA results.
Delete the first paragraph (lines 178 to 183, are not results).
Delete the second paragraph (lines 184-186) in this section and include it in Material and Methods section, Data Analyses subsection.
Line 187. S. perfoliatum in italics
The paragraph in lines 194-196 is not clear.
Table 1. the organ names in plural. Include ANOVA test results. Means with different letters (a, b, c) are significant at p<0.05 (Please mention statistical test used). Delete the raw HSD5%.
Figure 1. Means 243 (± SE) with different letters are significant at p<0.05 (mention statistical test used). Delete HSD5%=91
In “2.2.1. The free proline content subsection” delete the first paragraph (lines 248-254).
Table 4. Include the ANOVA results and indicate if the Tukey test was used. Organ names in plural.
In figure 4 include the PC2 (%)
Discussion
Line 458. Exchange Silphium perfoliatum L. (Asteraceae) for S. perfoliatum
In line 469 delete the period in “cells.” word.
In line 498 delete the period in “studies.” word.
In line 560 include the genus in: E. nigrum leaves [119], S. oleracea [120], P. vulgaris
In line 591 the Brassicaceae family without italics
Materials and Methods
In line 618 include the period in “etc” word, and deleted “4.2.”
In “4.5.1. Determination of the heavy metal‘s accumulation in plant organs”, please explain the acid digestion method used” and the meaning of ASS. Also, describe how many plant tissue was used.
In line 701 delete the year in the Bates et al. reference.
Reviewer 2 Report
Heavy metals are well-known environmental pollutants due to their toxicity, persistence in the environment, and bioaccumulative nature. Submitted report is focused on two main issues:
1) the phytoremediation capacity of the Silphium perfoliatum L. species by growing plants under floating hydroponic systems on nutrient solutions with Cu-400 ppm, Zn -1200 ppm, Cd-20 ppm, Pb-400 ppm, Cu+Zn+Cd+Pb (400/1200/20/400)
2) the effects of those heavy metals on the proline content, photosynthetic pigments (chlorophyll a and b, carotenoids) and enzymatic activity of superoxide dismutase (SOD), catalase (CAT) and peroxidase (POD) in different organs of S. perfoliatum plants.
Comments:
Introduction
This chapter is too ample and it should be more concise. Paragraphs that are not directly related to the topic of the work should be deleted (e.g. lines 64-68; 91-102) or shortened (e.g. 103-127, 131-155).
Material and methods
Plant metabolism is not only modified by cations, but also by anions. Why different metal salts: CuSO4, Cd(CH3COO)2 ·2H2O, Pb(CH3COO)2 ·3H2O, ZnSO4 ·7H2O. were added into the Hoagland medium?
What was the reason that the following concentrations of Cu-400 ppm, Zn -1200 ppm, Cd-20 ppm, Pb-400 ppm, Cu+Zn+Cd+Pb (400/1200/20/400) were chosen for the experiment? What was the influence of those metal concentrations on plant growth and morphology after 20 days of exposure to heavy metals?
The chemical reaction between Pb2+ and PO4-3 (e. g. coming from dissociation of NH4H2PO4) produces Pb3(PO4)2 that is insoluble in water. How lead phosphate precipitation affected the Pb2+ concentration in the Hoagland medium?
Results
Information, previously published by other authors, should be deleted from this chapter and included into the introduction (lines 179-183; 248-254; 268-271; 329-333).
Tab. 1 Cadmium and lead were not added to the control nutrient solution. To what the presence of these metals in untreated plants can be attributed?
Lines: 248-266
In which organ (root, stem and/or leave) was the level of proline determined? This is not obvious from the text and the caption of the figure 2.
Discussion
Lines 165-171 “Silphium perfoliatum L. (cup plant) appears to be one of the indicated species for the remediation of ecosystems subject to heavy metal pollution…”
According to chapters 2.1. and 3.1 the accumulation of heavy metals was most pronounced in the stem (Cu and Zn), stem and root (Cd) and root (Pb), regardless the method of metal application (one selected metal or simultaneously four metals). What is the conclusion coming from these results? What is phytoextraction potential of S. perfoliatum (a terrestrial plant), for removing Cu, Zn, Cd and Pb from polluted soil? According to the text in lines 652-653 “Over the last few decades, comparative studies have been performed to prove that floating hydroponic systems and soil are interchangeable techniques…”.
Lines 607-609 “The research results show that S. perfoliatum plants tried to protect against heavy metal exposure by activating the antioxidant enzyme system, and the level of activity seems to be directly dependent on the type of heavy metal, concentration and exposure time”
What is the novelty of these results apart from the employment of S. perfoliatum as a model plant?
Conclusions
Line 770 “Free proline biosynthesis is more intense…”. Actually proline biosynthesis was not examined. It would be more precise to use proline level or accumulation.
General comments
The response of plants to heavy metals have been studied for several decades. It is generally accepted that excess of metal ions, among others, influence proline accumulation, chlorophyll and carotenoids amount and composition, activity of antioxidant enzymes such as superoxide dismutase (SOD), peroxidase (POD) and catalase (CAT). It has been proved that plant response depends on the species, organ, metal type, concentration, and time of exposure to stress. To date all these aspects have been extensively described and discussed. In my opinion, the submitted manuscript confirms rather than expands the current state of knowledge in this area. Low originality/novelty of the results is an additional reason due to I am unable to recommend submitted manuscript for publication in Plant.
The effects of high concentrations of heavy metals on some physiological and biochemical parameters of the S. perfoliatum growing in floating hydroponic systems in nutrient solutions with excess of heavy metals (Cu; Zn; Cd and Pb) have not been studied so far. Therefore I believe that these results can be considered for publication in other plant journals.
Reviewer 3 Report
Dear Editor:
Thank you for giving me the opportunity to revise the MS entitled “The Impact of Heavy Metal Accumulation on Some Physiological Parameters in Silphium perfoliatum L. Plants Grown in Hydroponic Systems” by SUMALAN and his/her colleagues that was submitted to “plants”. The MS submitted is suitable for plants, and some interesting results were showed. However, there are several requirements that have to consider by the authors. In this regard, the following comments are requested to be addressed by the authors:
1. The Abstract section needs to be carefully revised.
2. I think the introduction section should be revised better. Some paragraphs and sentences can be merged and refined. In addition, the introduction section is too lengthy.
3. The novelty of this literature research should be inserted in the text clearly.
4. The discussion section can be more in-depth.
5. The contents can be simplified and consolidated in the Materials and Methods section. Please check carefully. I think 4.2 is redundant in the sentence “……and tolerance to stress, etc [50; 51 60; 61; 134].4.2.”
6. There are many errors in the full text and supplementary, which need to be carefully checked.
7. Please note that the format of references must be uniform.
I would suggest that the authors review and include the following recent studies about bioremediation and phytoremediation to improve the manuscript.
Su, R.; Ou, Q.; et.al., Comparison of phytoremediation potential of Nerium indicum with inorganic modifier calcium carbonate and organic modifier mushroom residue to lead-zinc tailings. Int. J. Environ. Res. Public Health 2022, 19, (16), 10353.
Su, R.; Ou, Q.; et.al., Organic–inorganic composite modifiers enhance restoration potential of Nerium oleander L. to lead–zinc tailing: application of phytoremediation. Environ Sci Pollut R. 2023, DOI:10.1007/s11356-023-26359-w.
Best regards,
Round 2
Reviewer 2 Report
Thank you very much for replying to my questions and comments.
One more note:
Line 701-702 The lowest values of metal accumulation were observed in organs exposed to Pb.
Shouldn't this sentence be related to POD activity, not to metal accumulation?
Reviewer 3 Report
The manuscript has been sufficiently improved to warrant publication in Plants.